# SMARTPLAY : A BENCHMARK FOR LLMS AS INTELLIGENT AGENTS

**Yue Wu**[12,*]**, Xuan Tang**[1]**, Tom Mitchell**[1]**, Yuanzhi Li**[12]
[1]Carnegie Mellon University, [2]Microsoft Research

## ABSTRACT

Recent large language models (LLMs) have demonstrated great potential toward intelligent agents and next-gen automation, but there currently lacks a systematic benchmark for evaluating LLMs' abilities as agents. We introduce SmartPlay: both a challenging benchmark and a methodology for evaluating LLMs as agents. SmartPlay consists of 6 different games, including Rock-Paper-Scissors, Tower of Hanoi, Minecraft. Each game features a unique setting, providing up to 20 evaluation settings and infinite environment variations. Each game in SmartPlay uniquely challenges a subset of 9 important capabilities of an intelligent LLM agent, including reasoning with object dependencies, planning ahead, spatial reasoning, learning from history, and understanding randomness. The distinction between the set of capabilities each game test allows us to analyze each capability separately. SmartPlay serves not only as a rigorous testing ground for evaluating the overall performance of LLM agents but also as a road-map for identifying gaps in current methodologies. We release our benchmark at github.com/microsoft/SmartPlay.

## 1 INTRODUCTION

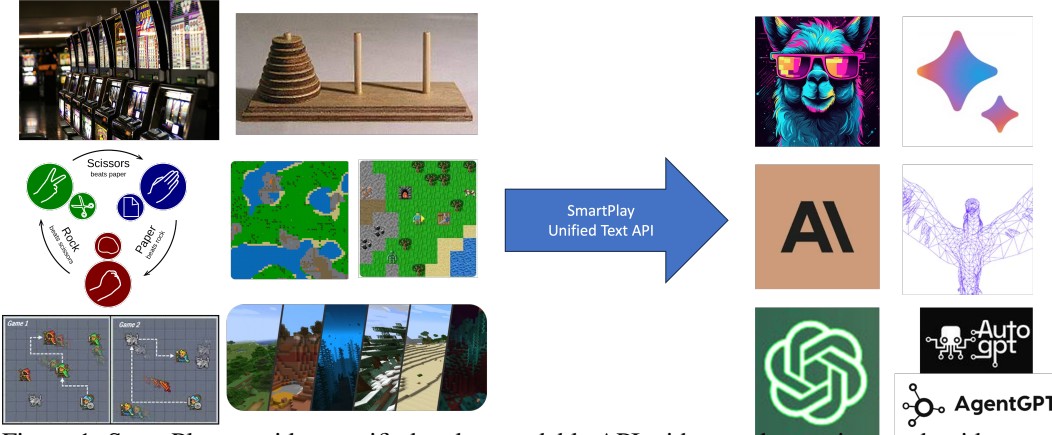

Figure 1: SmartPlay provides a unified and expandable API with text observations and guidance to perform turn by turn LLM inference on Two-armed Bandits, Rock Paper Scissors, Messenger (Hanjie et al., 2021), Crafter (Hafner, 2021), and Minecraft (Fan et al., 2022) creative navigation tasks.

Creating intelligent agents (Wooldridge & Jennings, 1995), that *perceives* its environment and perform autonomous *actions*, has been one of the core objectives of A.I. (Laird et al., 1987; Russell, 2010) Recently, large language models (LLMs) (Smith et al., 2022; Chowdhery et al., 2022; OpenAI, 2023; Manyika; Driess et al., 2023; Touvron et al., 2023) have made remarkable progress in various tasks (Bubeck et al., 2023). Some language models demonstrate exceptional planning (Ahn et al., 2022; Wu et al., 2023b), reasoning (Wu et al., 2023a; Shinn et al., 2023), and problem-solving (Madaan et al., 2023; Kim et al., 2023) abilities, enabling the potential as generalist agents for virtual-reality (Park et al., 2023) or real-world problem-solving.

Such potential has attracted strong interest on applications where LLM systems actively invoke tools and APIs to complete a wide range of tasks goals (Significant-Gravitas; Yoheinakajima; Reworkd;

---

*Work done during internship at Microsoft Research. Correspondence to ywu5@andrew.cmu.edu

Wang et al., 2023a; Qin et al., 2023), and actively interact and make changes in an environment to achieve specific results (Wang et al., 2023b;a; Wu et al., 2023b;c). LLMs as agents could be seen as an important step toward next-gen automation.

Despite great public attention, the capabilities of LLMs as agents have not been systematically studied, partly due to the lack of standardized LLM benchmark for agent-environment interaction. Current LLM benchmarks have been designed for static knowledge and reasoning (Hendrycks et al., 2020; Liang et al., 2022; Srivastava et al., 2022a; Zhong et al., 2023), or helpful and harmless conversations (Bai et al., 2022; Zheng et al., 2023a; Dubois et al., 2023), overlooking applications to intelligent agents.

We note 4 key challenges for intelligent LLM agents not captured in previous benchmarks. First, lots of real-world tasks require an agent to do long-horizon **planning**. Second, many events are probabilistic and an intelligent agent is expected to **understand the odds**. Third, an intelligent agent needs **spatial reasoning** to understand our 3D world. Fourth, when encountered with unseen situations, an intelligent agent should be able to **learn from interactions or mistakes**.

On the other hand, games have long been identified as go-to benchmarks for intelligent generalist agents (Pell, 2011; Genesereth et al., 2005; Whiteson et al., 2010; Schaul et al., 2011; Bellemare et al., 2013; Côté et al., 2019; Hafner, 2021; Guss et al., 2021; Fan et al., 2022). At the core of game design (Koster, 2013), successful games often involve "problem-solving", "calculation of odds", "spatial reasoning", "changing difficulties", and "well-defined and quantifiable outcome", therefore offering perfect complement to existing LLM benchmarks. Finally, some game environments are procedurally generated and game states grow exponentially, making games more robust against evaluation dataset contamination as observed in recent works (Touvron et al., 2023). Experimentally, we observe LLMs struggle to memoize intermediate states of a simple 3-disk Tower of Hanoi game.

Taking a unique agent perspective in benchmarking LLMs, we introduce SmartPlay, a benchmark from 6 distinct games augmented with language descriptors for visual observation (Figure 1), offering up to 20 different settings and infinite environment variations. Each game presents unique challenges that span multiple dimensions of intelligent agents, as detailed in Table 3. The games range in complexity, from requiring simple one-step reasoning and rule-following in Bandits, to intricate long-term planning, multi-hop dependencies, and learning from interactions in Crafter (Hafner, 2021) and Hanoi. SmartPlay engages LLM agents in both deterministic and stochastic settings, demanding skills from basic text understanding to 3D spatial reasoning.

Games in SmartPlay have been built with well-defined objectives and evaluation metrics: completion rate, reward, score. Therefore, SmartPlay provides a fully automated pipeline to conduct standardized evaluation for LLMs.We use SmartPlay to compare the agent performance of recent LLMs, and identify several research gaps for applying LLMs as agents. We believe that SmartPlay sets a goal that is reachable in a short time-frame yet formidable to require new breakthroughs.

## 2 Capabilities Necessary for Intelligent Agents

Borrowing concepts from game design (Koster, 2013), we identify 9 key abilities important for intelligent LLM agents, and identify multiple degrees for each capability:

a) **Long text understanding**: general LLM capability.
   - We define 4 degrees based on document length and syntactic variations: 1) few fixed lines, 2) few fixed paragraphs, 3) with syntactic variations, 4) and longer than 1 page (500 words).
b) **Reasoning**: multi-hop logical reasoning and deduction, often required for analyzing the interactions of in-game objects or action conditions/dependencies.
   - We define 3 degrees based on reasoning hops: 1) $(0 \sim 1)$, 2) $(2 \sim 3)$, 3) $(> 3)$.
c) **Instruction/Rule following**: follow rules and instructions set by environment or users.
   - We define 3 degrees based on number of game rules: 1) single rule, 2) $(< 5)$, 3) $(5+)$
d) **Planning**: long-horizon in-context planning to achieve a complex goal.
   - We define 3 degrees based on planning steps, and concurrent objectives which requires goal prioritization: 1) $< 5$ planning steps, 2) $5+$ planning steps, 3) concurrent objectives
e) **Generalization**: Excels at a wide range of tasks.

- • We define 3 degrees based on the variability the game provides: 1) fixed environment, 2) fixed game word with random objectives, 3) procedurally generated game world

**f) Understanding the odds**: analyze and estimate the probability of random events.

- • We define 3 degrees based on the importance randomness in the environment: 1) no randomness, 2) randomness present in game, 3) randomness as core game mechanism

**g) Learning from interactions**: acquire environment knowledge from live interactions.

- • We define 4 degrees based on the number of unique interactions to learn from: 1) no learning required, 2) single interaction, 3) $< 5$ interactions, 4) $5+$ interactions

**h) Error/Mistake handling**: recover from mistakes (e.g., correcting from erroneous trajectory).

- • We define 3 degrees based on if mistake handling may be necessary and if additional reasoning and re-planning is necessary: 1) not required, 2) simple rollback corrects error, 3) reasoning and re-planning required to correct error.

**i) Spatial reasoning**: understand our world in 2D/3D. Spatial reasoning is typically required to understand directions and navigate through the game world (e.g., navigating the 2D/3D world).

- (a) We define 3 degrees based on dimensionality: 1) $0 \sim 1D$, 2) 2D, 3) 3D

## 3 GAMES IN SMARTPLAY

### 3.1 RESEARCH CHALLENGES

The SmartPlay benchmark encapsulates a diverse set of challenges that evaluate various AI capabilities, as itemized in Figure 2. For instance, Bandits primarily focuses on understanding the odds, requiring minimum text understanding and rule-following. On the other hand, Rock Paper Scissors uniquely puts an emphasis on understanding the odds and multiple game rules. Hanoi presents an advanced setting for object dependency reasoning, strategic planning, and handling mistakes. Messenger puts challenge on 2D spatial reasoning, reading syntactic variations and conducting multi-hop reasoning. Meanwhile, Minecraft offers a unique challenge in 3D spatial reasoning and generalization within a procedurally generated world. We hope the SmartPlay benchmark would serve as a tool for identifying these nuanced gaps and directing future research.

While each game poses its unique challenges, the SmartPlay benchmark also evaluates an agent's capability to integrate these skills. For example, Crafter stands as the most comprehensive testbed, combining long texts, multiple interactions, concurrent objectives, and error handling into a single environment. Crafter highlight the need for future research to focus not just on isolated skills, but also on combining these skills into a unified, adaptive agent.

### 3.2 TWO ARMED BANDITS

The two armed bandit benchmark is inspired by popular implementations[1] of bandit problems.

The LLM agent is provided two slot machines with hidden pre-defined reward probabilities $p_1, p_2$. For slot machine $i$, the reward for the two possible out-comes are: $r_i$ for pay-off event and $-r_i$ for no-pay-off event. The goal of the game is to find the arm with better return and maximize the reward over the course of 50 rounds. The human written manual informs the LLM of the number of slot machines (two) and the objective.

An agent must keep track of win/losses from its past roll-out and balance exploration across the two slot machines vs. exploitation of the more rewarding one. Overall, the challenges include: 1) long context understanding, 2) understanding randomness, 3) learning from interactions.

To prevent game exploitation caused by biased actions, we randomize the score and probabilities for each action by shuffling the order of the paired list: $[(p_1, r_1), (p_2, r_2)]$.

### 3.3 ROCK PAPER SCISSORS

Same rules as the famous zero-sum game Rock Paper Scissors[2].

---

[1] github.com/JKCooper2/gym-bandits

[2] wikipedia.org/wiki/Rock_paper_scissors

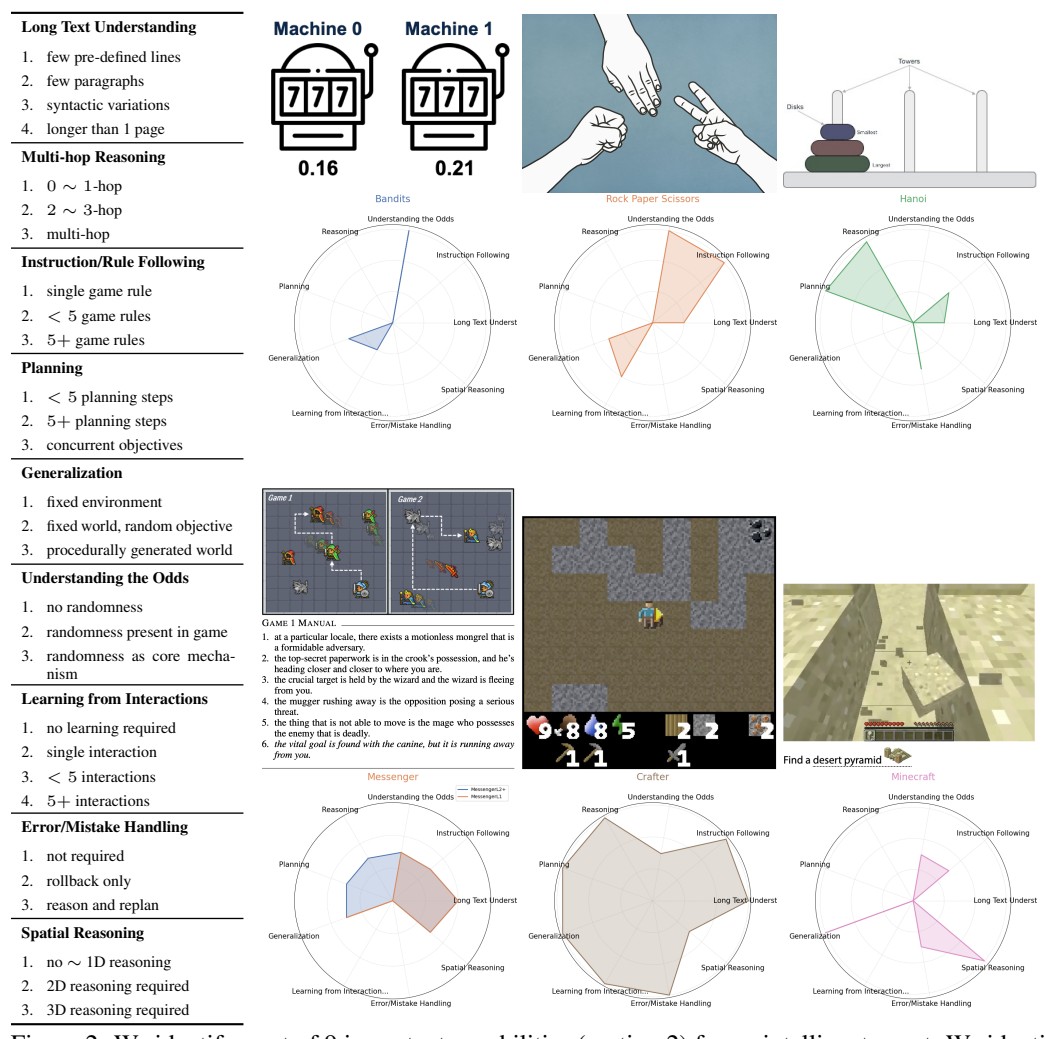

Figure 2: We identify a set of 9 important capabilities (section 2) for an intelligent agent. We identify different degrees of challenge for each capability as shown on the left. Each game in SmartPlay challenges unique set of capabilities at different degrees, as shown in the spider charts. We include numerical values of the spider plots in Table 3.

The LLM agent plays against a hand-coded opponent that follows a hidden pre-defined strategy with probabilities $p_1, p_2, p_3$ for rock, paper, and scissors respectively. The scores for winning under each action is pre-defined and revealed to the LLM as $s_1, s_2, s_3$. The human written manual provides instruction on the possible actions and how the win/draw/lose of each round is calculated.

An agent must keep track of win/losses from its past roll-outs to analyze the opponent behavior, and then exploit the opponent to maximize payoff. Overall, the challenges include: 1) long text understanding, 2) understanding the odds, 3) learning from interactions, 4) instruction following.

To prevent game exploitation caused by biased actions, we randomize the score and probabilities for each action by shuffling the order of the paired list: $[(p_1, s_1), (p_2, s_2), (p_3, s_3)]$.

## 3.4 TOWER OF HANOI

The Tower of Hanoi[3] is a classic puzzle game that challenges the player to move a stack of disks from one rod to another, using a third rod as an auxiliary. The game has two rules: only one disk can be moved at a time, and a larger disk cannot be placed on top of a smaller one.

The goal of the game is to move all the disks from the first rod to the last one in the minimum number of moves, and the game can be solved using a recursive algorithm that follows these steps:

---

[3]github.com/RobertTLange/gym-hanoi/tree/master

1. Move n - 1 disks from the source rod to the auxiliary rod, using the destination rod as an intermediate.
2. Move the largest disk from the source rod to the destination rod.
3. Move n - 1 disks from the auxiliary rod to the destination rod, using the source rod as an intermediate.

The human written manual contains a description of the game set-up and allowed actions. In addition, we also include an example illustration of the starting and goal configuration, alongside an example of allowed/disallowed moves.

The Tower of Hanoi requires the agent to think strategically and plan ahead, and put strict requirements on the LLM agents' ability to understand and follow the rules of the game. The game can become more challenging if the agent makes a mistake. Sometimes, an agent may have to undo several moves to correct an error. Overall, the challenges include: 1) planing, 2) reasoning, 3) error handling.

## 3.5 MESSENGER

MESSENGER (Hanjie et al., 2021) which features multiple game variants with procedurally generated game dynamics and accompanying text manuals. The overall game mechanics of MESSENGER involve obtaining a message and delivering it to a goal. The benchmark is shipped with 3 levels of difficulties (referred as stages in Hanjie et al. (2021)).

To succeed in MESSENGER, an agent must first relate entities and dynamics of the environment to their reference synonyms in the manual, identify message and goal objects, and navigate to bring the message to the goal while avoiding the enemy. The manual, by design, is challenging to understand even for human readers. Level 1 primarily challenges the agent's 1) long text understanding and 2) generalization . Level 2 includes additional challenge on the agent's 3) reasoning, and 4) 2D spatial reasoning. Level 3 increases difficulty by adding distraction objects.

The original manuals provided by Hanjie et al. (2021) contain descriptions of the entities and world dynamics obtained through crowd-sourced human writers. We augment the manual with a specification on the game objective, and an "advice" for LLM agent to first identify goal objects and then approach its objective. The "advice" reduces the difficulty of the hard-to-parse manual.

## 3.6 CRAFTER

The Crafter environment (Hafner, 2021) is a procedurally generated, open-world survival game designed to test RL algorithms. Inspired by Minecraft, it features a grid-world with top-down observation and a discrete action space of 17. The game includes 22 achievements in a tech-tree of depth 7 and provides information on the player's health, food, water, rest levels, and inventory. Crafter captures many of Minecraft's key research challenges, offering a more streamlined and faster environment for conducting experiments and gathering results.

We provide the "context" string from Wu et al. (2023c) as the manual, generated by parsing the LaTeX source-code of (Hafner, 2021). The "context" string has been shown to greatly improve performance of GPT-4 and text-davinci-003 on Crafter (Wu et al., 2023c).

To succeed in Crafter an LLM agent has to first understand and master a variety of reusable skills composed of 17 actions. The agent needs to learn to navigate through up to 5 2D terrains (biomes), avoiding obstacles and dangerous creatures. The agent also needs to collect different resources and craft more advanced weapons/tools to unlock more skills and achievements, while at the same time balance crafting goals with survival goals like maintaining health, thirst, food, and rest (Hafner, 2021). Overall, the challenges include: 1) 2D spatial reasoning, 2) mistake handling, 3) long text understanding, 4) planning, 5) generalization, 6) correcting from mistakes. Interestingly, the "context" string does not capture all information necessary to succeed in the game, i.e., it requires 2 woods to craft the crafting table, and 8 stones to craft the furnace. The agent has to 7) learn from interaction.

## 3.7 MINECRAFT

Minecraft is one of the most popular games in history[4]. The game world is virtually infinite and procedurally generated. The game observation is composed of rough 3D objects representing various materials, such as dirt, stone, ores, tree trunks, water, and lava. Minecraft has been widely studied as

---

[4]wikipedia.org/wiki/Minecraft

a benchmark for intelligent multi-tasking agents (Guss et al., 2021; Fan et al., 2022; Hafner et al., 2023; Yuan et al., 2023; Wang et al., 2023b;a). However, due to the fact that most current LLMs do not have vision capabilities, we simplify the Minecraft benchmark (Fan et al., 2022) and only consider a small set of creative tasks where the primary objective is to find specific biomes, so an LLM could control a hand-coded agent to perform navigation in the 3D world.

For the human written instruction manual, we inform the agent that its goal is to find a specific biome $g$ in Minecraft, and offer an advice on how to interpret the visual descriptor output for Minecraft.

To succeed in the creative "find" tasks, a LLM agent has to have enough domain knowledge about different biomes in Minecraft, and be able to correlate visual observation (text description of visual world) with domain knowledge, and navigate in a 3D environment. Overall, the challenges include: 1) planning, 2) *domain knowledge*, 3) 3D spatial reasoning, 4) generalization.

## 4 Using SmartPlay

### 4.1 Environment Interface and Evaluation Protocol

| Env | Input | Manual | History | Rollout | Action Space | Trials |
|---|---|---|---|---|---|---|
| Bandits | Text | Background | 50 | 50 | 2 | 20 |
| RockPaperScissors | Text | Background,Rules | 50 | 50 | 3 | 20 |
| Hanoi | Text | Background,Rules,Examples | 30 | 30 | 6 | 10 |
| Messenger | Visual description | Background,Rules,Advice | 2 | 4∼128 | 5 | 100 |
| Crafter | Visual description | Background,Rules,Advice | 5 | 10k | 17 | 10 |
| Minecraft | Visual description | Objective | 2 | 200 | 4 | 20 |

Table 1: Specifications for each game in SmartPlay. In addition to the table, the manual input contains a list available actions for all games. Input, manual, action space, and rollout length should not be modified. History length and trial numbers could be increased to suite future needs.

For ease of use and wide compatibility, SmartPlay follows a unified OpenAI Gym interface (Brockman et al., 2016) for all games, with text-based observations, text-based manuals with content as described in Table 1, text describing historical actions and observations covering past steps of length "history length", and flat categorical actions. Due to the randomness in some games, we recommend running each game multiple times and reporting the average metrics.

Input, manual, action space, rollout length (the maximum environment steps allowed by each game), and trial numbers for each game are specified in Table 1. These settings are fixed and should not be modified. However, future research may require longer history length or more trials for some games. These parameters can be adjusted to suit specific needs, but the changes should be explicitly stated. We provide recommended values (also used in our experiments) for the parameters in Table 1.

For completeness, we provide example inputs for each game in Appendix C. Note that all directions in SmartPlay are described in "east, south, west, north, above, below" In the actual gameplay, SmartPlay API also includes a list of actions for the LLM agent to pick from.

### 4.2 Evaluation Metrics

We define three metrics: reward, completion rate, score. To ensure compatibility with prior works, **reward** aligns with the score/reward definition in games originally designed for RL (i.e., Bandits, Rock Paper Scissors, Messenger, Crafter (Hanjie et al., 2021; Hafner, 2021)). **Completion rate** measures the rate of successful completion for games with quantifiable objectives (i.e., Hanoi, Messenger, Minecraft). Finally, we introduce **score** for every game in the benchmark to provide a summary of performance. For Bandits and Rock Paper Scissors, the score is defined the number of times the LLM action matches the environment optimal action; for Hanoi, the score is defined as the number of disks successfully moved to the goal peg; for Messenger, the score is the same as the reward (Hanjie et al., 2021) of each round of game; for Crafter, the score is defined as the number of unlocked achievements at every step, summed across the whole game; for Minecraft, the score is defined as the indicator of whether the "find" objective for the game has been completed.

## 5 EXPERIMENTAL RESULTS

Using the SmartPlay API, we follow Wu et al. (2023c) and directly prompt an LLM: "What is the next action to take, let's think step by step.", with manual, history, and current observation as context. We then query the LLM: "Choose the best executable action from the list of all actions. Write the exact chosen action." for an answer directly mapped to one of the environment actions.

### 5.1 QUANTITATIVE ANALYSIS

| LLM | Bandit | RPS | Hanoi | MessengerL1 | MessengerL2 | Crafter | Minecraft |
|---|---|---|---|---|---|---|---|
| Human Baseline | 1.00 | 1.00 | 1.00 | 1.00 | 1.00 | 1.00 | 1.00 |
| GPT-4-0613 | 1.00 | 0.91 | 0.83 | 0.90 | 0.93 | 0.26 | 0.61 |
| GPT-4-0314 | 0.97 | 0.98 | 0.90 | 0.87 | 0.97 | 0.32 | 0.59 |
| text-davinci-003 | 1.04 | 0.40 | 0.50 | 0.62 | 0.46 | 0.07 | 0.45 |
| Claude | 0.72 | 0.47 | 0.67 | 0.44 | 0.60 | 0.05 | 0.50 |
| Bard | 0.86 | 0.30 | 0.67 | 0.61 | 0.40 | 0.04 | 0.54 |
| llama-2-13b | 0.50 | 0.35 | 0.37 | 0.12 | 0.13 | 0.04 | 0.61 |
| llama-13b | 0.68 | 0.50 | 0.33 | 0.16 | 0.06 | 0.04 | 0.50 |
| vicuna-13b | 0.64 | 0.17 | 0.07 | 0.00 | 0.12 | 0.02 | 0.43 |

Table 2: Comparison of performance of different LLMs in terms of average score on BanditTwoArmedHighLowFixed-v0, RockPaperScissorBasic-v0, Hanoi3Disk-v0, MessengerL1-v0, MessengerL2-v0, Crafter-v0, MinedojoCreative0-v0. All scores are normalized relative to human performance (unnormalized version in Table 4). GPT-4 variants out-perform other LLMs by significant margins, but still greatly under-perform human baselines. We observe significant performance gaps between SOTA LLMs and human baseline on Hanoi, Crafter, and Minecraft. Hanoi, Crafter challenges planning and reasoning, and Minecraft challenges 3D spatial reasoning.

To reduce the cost of queries, we pick 7 settings that requires a minimal experimentation but provides comprehensive coverage over important agent capabilities. We experiment with 9 recent popular open-source and proprietary LLMs and report the average score in Table 2.

Overall, GPT-4 variants significantly out performs other proprietary models, which outperform open-source models by significant margins.

**There is still significant room for improvement for LLM as agents:** Despite the impressive performance of GPT-4 variants, there is still a significant gap between GPT-4 and human baseline performance on more challenging benchmarks, with a 10% gap on 3DiskHanoi, 40% on Minecraft creative tasks, and 70% on Crafter.

**Other proprietary LLMs struggle to keep up with GPT-4:** We observe a more than 20% gap between GPT-4 and other proprietary models like Claude, Bard, and text-davinci-003 across all games except Minecraft. Furthermore, on comprehensive benchmarks like Crafter, GPT-4 variants achieves 3 times higher scores than other proprietary models.

**Open-source LLMs have a long way to go:** Open-source LLMs achieves less than half the performance of GPT-4 variants on simple Bandit and Rock-Paper-Scissors tasks, and 1/8 the performance on more challenging tasks. The fine-tuned Vicuna-13b model performs much worse than the base LLAMA-13b.

**3D Spatial reasoning remains a challenge for LLMs:** The Minecraft benchmark appears equally challenging to all LLMs due to its unique requirement for 3D spatial reasoning. All LLMs behave similarly in Minecraft creative tasks, with the best model at 60% of human baseline performance.

To offer additional insights into the individual agent capabilities of LLMs as identified in Figure 2, we compute, for each capability $c$, the capability score $p^c_{LLM}$ of an LLM as the average of human normalized score $s_g$ over each game $g$, weighted by the degree $d^g_c$ at game $g$ presents challenge $c$: $p^c_{LLM} = \frac{\sum_g d^g_c s_g}{\sum_g d^g_c}$. We plot the capability scores into 3 groups in Figure 3: GPT-4 variants, other proprietary models, and open-source models.

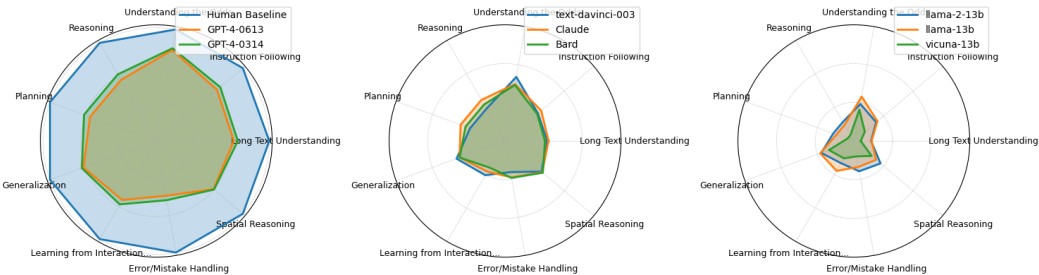

Figure 3: **Left:** comparing the two GPT-4 variants with Human Baseline performance as reference. **Middle:** comparing text-davinci-003, Claude, and Bard. **Right:** comparing open-source llama-2-13b, llama-13b, vicuna-13b models.

The two GPT-4 variants perform similarly overall, with GPT-0614 doing slightly worse on planning and reasoning. We also identify that GPT-4 variants score lower on learning from interactions, error/mistake handling, and spatial reasoning.

Claude demonstrates overall better performance than Bard, especially in planning, reasoning, instruction following. Compared to the other two proprietary models, text-davinci-003 appears biased toward learning from interaction and randomness, and is particularly weaker at instruction following, planning and reasoning.

LLAMA-2-13b and LLAMA-1-13b performs similar on the high level, with LLAMA-2-13b performing better at planning, reasoning, and error handling, but worse in learning from randomness and interactions. Vicuna-13b loses a lot of reasoning, planning, long text understanding, and error/mistake handling capabilities after fine-tuning.

## 5.2 QUALITATIVE ANALYSIS

**Learning from interactions:** In Bandits and Rock Paper Scissors, proprietary LLMs demonstrate promising potential for learning from history and interactions. We observe the agents first following a exploratory strategy and then exploiting the biased opponent based on the past observations. In Crafter, GPT-4 variants consistently attempts to build crafting table with 1 wood and recovers from the failure to build crafting table with 2 woods.

**Data/environment contamination:** For the Tower of Hanoi, it is expected that the LLMs have been trained on the exact same problem. Surprisingly, although all LLMs are able to provide the solution at the starting configuration where all disks are on the first rod (some may even write out the recurrence for the solution), most LLMs could not solve the problem and gets confused quickly after a few moves, where the disks are distributed over all three rods. We suspect that this is due to the intermediate states do not appear often in the LLM's training sets. Such observation verifies our belief that games could be more robust to dataset contamination.

**Spatial Reasoning:** We observe that LLMs often have a bad sense of spatial locations and struggle with navigating to new locations. For example, in Minecraft, we often observe LLMs often take moves that are contradictory over time, i.e., a bunch of "move north" followed by a bunch of "move south", undoing a lot of its own efforts at exploration.

## 6 RELATED WORKS

### 6.1 LLM EVALUATION

The task of evaluating LLM performance has become increasingly challenging given the rapid progression of LLMs. Generalist benchmarks usually employ a wide range of tasks and languages to test general knowledge and reasoning (Hendrycks et al., 2020; Liang et al., 2022; Srivastava et al., 2022a; Zhong et al., 2023), where small language models are getting close performance compared to the state-of-the-art large language models Li et al. (2023); Gunasekar et al. (2023); Eldan & Li (2023). However, those benchmarks struggle to cover interaction styles like instruction following Ziegler et al. (2019) or conversations Bai et al. (2022). The go-to approach for evaluating LLM for conversation

is pairwise model comparison, which performs pair-wise comparison of output of the LLM and a reference LLMs to produce a ranking (Zheng et al., 2023b). The ranking was originally performed by human, but could be automated with a significantly more powerful LLM (Chiang & Lee, 2023; Zheng et al., 2023a; Dubois et al., 2023). However, such evaluation techniques depend on an expert model or human who can reliably compare the performance of different LLMs, which limits the application to SOTA LLMs like Claude-2 or GPT-4. Moreover, existing benchmarks fail to capture key characteristics of intelligent agents like understanding of randomness, spatial reasoning, and error handling.

## 6.2 USING GAMES TO EVALUATE GENERALIST AGENTS

The idea of using games to evaluate the performance of agents has a long history in A.I. Pell (2011); Schaul et al. (2011); Whiteson et al. (2011) presented early ideas and motivation for using games to measure the general capabilities of an agent, and discussed challenges in measuring A.I. agent performance. A series of popular benchmarks (Brockman et al., 2016; Vinyals et al., 2017; Tunyasuvunakool et al., 2020) were created including Atari (Bellemare et al., 2013) and DeepMind lab (Beattie et al., 2016). As the capabilities of A.I. agents improve, researchers developed open-ended generalist games (Savva et al., 2019; Abramson et al., 2020; Hafner, 2021; Srivastava et al., 2022b) like NetHack (Küttler et al., 2020) or Minecraft (Guss et al., 2021; Fan et al., 2022).

SmartPlay takes a suite of benchmarks (Brockman et al., 2016; Hafner, 2021; Fan et al., 2022) developed over different times to best represent a broad range of difficulties and skills.

## 6.3 CREATING/CONVERTING TO TEXT GAMES

Text games (Côté et al., 2018; Küttler et al., 2020; Zhong et al., 2019; Hanjie et al., 2021) are interactive simulations where the game state and action space are in natural language, often used to benchmark skills like planning, exploration, and memory. SmartPlay features a text game (Messenger) with procedural game rule generation (Hanjie et al., 2021) to test the generalization of the LLM agents at language understanding and planning.

To capture real-world challenges like spatial-reasoning, we study converting 2D/3D games into text-games. Shridhar et al. (2020b) demonstrated the possibility of converting a 3D embodied indoor environment (Shridhar et al., 2020a) into a TextWorld (Côté et al., 2018) game by "listing" all the objects in text. However, such conversion relies on low-level controllers and teleportation, trivializing the environments for current LLMs (Micheli & Fleuret, 2021; Wu et al., 2023b). Therefore, we follow Wu et al. (2023c) to offer a list of objects/observations with directional relationship to the agent: "to your south-east." Such description allows LLMs to make meaningful progress without low-level controllers (Wu et al., 2023c).

## 7 CONCLUSION

In this work, we introduce SmartPlay, both a challenging benchmark and a methodology for evaluating LLMs' performance as agents. Our initial release of SmartPlay consists of Two-armed Bandits, Rock Paper Scissors, Messenger (Hanjie et al., 2021), Crafter (Hafner, 2021), and Minecraft (Fan et al., 2022) creative navigation tasks. SmartPlay benchmarks not only basic abilities like instruction following and in-context reasoning, but also evaluates capabilities like planning, understanding of randomness, 2D/3D spatial reasoning, and error handling, which are often underrepresented in existing LLM benchmarks. To achieve next-gen automation, we believe that language models should go beyond speaking fluent language (Eldan & Li, 2023), and become more intelligent agents that could interact with the world and human users. We hope that SmartPlay would catalyze research on building more capable and reliable LLM agents.

Finally, SmartPlay offers guidelines for easily adding games to the benchmarking suite. SmartPlay will be continuously improved to provide up-to-date challenges for next-gen LLMs.

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

## A    RESEARCH CHALLENGES

| Games | Bandits | Rock Paper Scissors | Hanoi | MessengerL2+ | Crafter | Minecraft |
|---|---|---|---|---|---|---|
| **Long text Understanding** | 1 | 2 | 2 | 3 | 4 | 1 |
| 1. few pre-defined lines | | | | | | |
| 2. few paragraphs | | | | | | |
| 3. syntactic variations | | | | | | |
| 4. longer than 1 page | | | | | | |
| **Reasoning** | 1 | 1 | 3 | 2 | 3 | 1 |
| 1. $0 \sim 1$-hop | | | | | | |
| 2. $2 \sim 3$-hop | | | | | | |
| 3. multi-hop | | | | | | |
| **Instruction/Rule Following** | 1 | 3 | 2 | 2 | 3 | 2 |
| 1. single game rule | | | | | | |
| 2. $< 5$ game rules | | | | | | |
| 3. $5+$ game rules | | | | | | |
| **Planning** | 1 | 1 | 3 | 2 | 3 | 1 |
| 1. $< 5$ planning steps | | | | | | |
| 2. $5+$ planning steps | | | | | | |
| 3. concurrent objectives | | | | | | |
| **Generalization** | 2 | 2 | 1 | 2 | 3 | 3 |
| 1. fixed environment | | | | | | |
| 2. fixed world, random objective | | | | | | |
| 3. procedurally generated world | | | | | | |
| **Understanding the Odds** | 3 | 3 | 1 | 2 | 2 | 2 |
| 1. no randomness | | | | | | |
| 2. randomness present in game | | | | | | |
| 3. randomness as core mechanism | | | | | | |
| **Learning from Interactions** | 2 | 3 | 1 | 1 | 4 | 1 |
| 1. no learning | | | | | | |
| 2. single interaction | | | | | | |
| 3. $< 5$ interactions | | | | | | |
| 4. $5+$ interactions | | | | | | |
| **Error/Mistake Handling** | 1 | 1 | 2 | 1 | 3 | 2 |
| 1. not required | | | | | | |
| 2. rollback only | | | | | | |
| 3. reason and re-plan | | | | | | |
| **Spatial Reasoning** | 1 | 1 | 1 | 2 | 2 | 3 |
| 1. 1D – no spatial reasoning | | | | | | |
| 2. 2D reasoning required | | | | | | |
| 3. 3D reasoning required | | | | | | |

Table 3: Research challenges associated with each of the 6 games. Since MessengerL1 does not cover multi-hop reasoning, only L2+ is included.

## B    MINECRAFT VISUAL DESCRIPTOR

The raw ground truth MineDojo (Fan et al., 2022) is a block level matrix (2D matrix for lidar rays and 3D matrix for surrounding blocks), which is very hard for human or LLMs to comprehend. Inspired by Wu et al. (2023c), we adopt a directional visual description scheme to encode the scene observation in text. Specifically, we first run connected component algorithm to group the same blocks that are connect into groups, and then describe the group's relative position to the agent based on the closest block from the group. For example, an observation in Figure 4 will be described as:

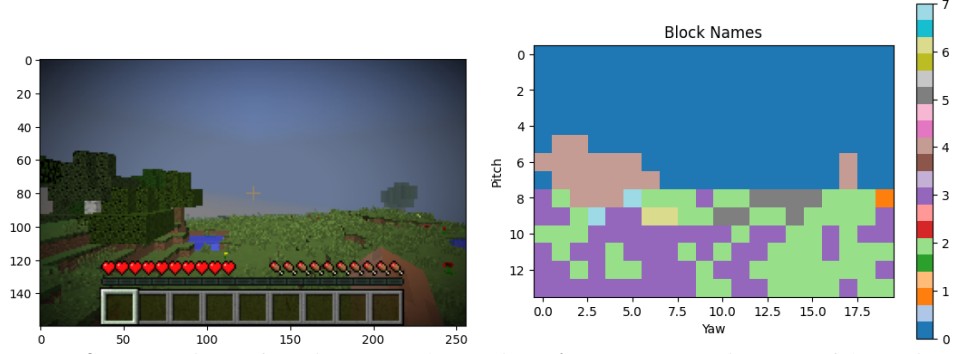

Figure 4: **Left:** Raw Minecraft environment observation **Right:** Segmentation map of the environment observation. as detected by lidar rays in MineDojo (Fan et al., 2022).

```
You see:
 - grass, 3.87 blocks away to your bottom, taking 38% of screen
 - plant, 22.10 blocks away to your bottom-right, taking 2% of screen
 - water, 33.88 blocks away to your bottom-left, taking 1% of screen
 - wood, 18.76 blocks away to your bottom-left, taking 0% of screen
 - flower, 32.98 blocks away to your right, taking 0% of screen
 - wood, 24.50 blocks away to your left, taking 0% of screen
 - leaves, 11.63 blocks away to your left, taking 6% of screen
 - leaves, 56.55 blocks away to your right, taking 1% of screen
```

## C  EXAMPLE INPUTS

### C.1  BANDITS

**Example input**

```
Instruction Manual:
You are in the casino with 2 slot machines in front of you.
Your goal is to try to earn the most from those slot machines.

Observation Example:
You pulled slot machine $i$, you received reward $r_i$.
```

### C.2  ROCK PAPER SCISSORS

**Example input**

```
Instruction Manual:
For the game Rock Paper Scissors, you and the opponent choose one of three options: rock, paper, or scissors.
After both players have chosen, the winner is determined as follows:
Rock crushes scissors (Rock wins, score $s_1$)
Scissors cut paper (Scissors win, score $s_2$)
Paper covers rock (Paper wins, score $s_3$)
If you lose, your score is the negative of the winner's score.
If both players choose the same option, it's a draw (score 0).
Your goal is to maximize your score.

Observation Example:
You chose $Rock$, and the opponent chose $Scissor$. You $won$ and received score $s_1$.
New round begins.
```

### C.3  HANOI

**Example input**

```
Instruction Manual:
The game consists of three rods (A,B,C) and a number of disks of various sizes, which can go onto any rod.
The game begins with the disks stacked on rod A in order of decreasing size, the smallest at the top (righthand side).
The objective is to move the entire stack to rod C, obeying the following rules:

 - Only one disk may be moved at a time.
 - Each move consists of taking the top disk from one of the stacks and placing it on top of another stack
or on an empty rod.
 - You cannot place a bigger disk on top of a smaller disk.
```

```
For example, considering movements from B under the following setting:
- A: |bottom, [0], top|
- B: |bottom, [1], top|
- C: |bottom, [2], top|
You are only allowed to move from B to C but not A, since the top of B (1) is smaller than the top of C (2)
but bigger than the top of A (0).

Finally, the starting configuration is:
- A: |bottom, [2,1,0], top|
- B: |bottom, [], top|
- C: |bottom, [], top|

and the goal configuration is:
- A: |bottom, [], top|
- B: |bottom, [], top|
- C: |bottom, [2,1,0], top|
with top on the right and bottom on the left

Observation Example:
You tried to move top disk of rod b to top of rod a. Current configuration:
- A: |bottom, [2, 1, 0], top|
- B: |bottom, [], top|
- C: |bottom, [], top|
```

## C.4  MESSENGER

### Example input

```
Instruction Manual:
In the game, MESSENGER, each entity can take on one of three roles: an enemy, message, or goal.
The agent's objective is to bring the message to the goal while avoiding the enemies.
If the agent encounters an enemy at any point in the game, or the goal without first obtaining the message,
it loses the game and obtains a reward of -1.

the dangerous enemy can be found next to the plane, which can not be moved.
you are being approached by a restricted document that is a robot.
the whale is the main objective.

To solve a game, you may find it helpful to list the objects that you see. Then for each object, match it with
an entity description, and identify whether it is good or bad to interact with the object.
The name specifications of in-game objects may not be exact matches. Please try identifying with synonyms.

Observation Example:
You took action Move South.

You (agent) don't have the message.

You see:
- airplane 7 steps to your south
- fish 13 steps to your south-east
- robot 5 steps to your south-east
```

## C.5  CRAFTER

### Example input

```
Instruction Manual:
Write all information helpful for the game in a numbered list.
1. Collect resources such as wood, stone, and iron to craft tools and weapons.
2. Build shelters to protect yourself from monsters at night.
3. Use tools and weapons to defend yourself against monsters.
4. Build bridges to cross lakes and rivers.
5. Dig tunnels to surprise monsters and outsmart them.
6. Plant saplings and defend them against monsters to ensure a steady food supply.
7. Eat Cow to restore health.
8. Collect Drink to restore thirst.
9. Place a Plant to eat for health.
10. Make a Wood Pickaxe to collect Stone.
11. Make a Wood Sword to defeat Zombies.
12. Make a Stone Pickaxe to collect Iron.
13. Make a Stone Sword to defeat Skeletons.
14. Place a Furnace to smelt Iron.
15. Collect Coal to smelt Iron.
16. Collect Iron to make an Iron Pickaxe and Sword.
17. Make an Iron Pickaxe to collect Diamond.
18. Make an Iron Sword to defeat Zombies and Skeletons.
19. Collect Diamond to progress further.
20. Unlock achievements to receive rewards.
```

21. Wake Up to start the episode.

In plain text. List all objects I need to interact/avoid to survive in the game.
Use "I would like to X object Y" in each step. Replace Y by the actual object, X by the actual interaction.
I would like to avoid zombies, skeletons, and spiders.
I would like to collect saplings.
I would like to craft a wood pickaxe.
I would like to collect wood.
I would like to craft a stone pickaxe.
I would like to collect stone.
I would like to craft a furnace.
I would like to collect coal.
I would like to collect iron.
I would like to craft an iron pickaxe.
I would like to collect diamonds.
I would like to craft an iron sword.
I would like to chase cows.
I would like to grow fruits.
I would like to drink from a lake.
I would like to sleep in a safe place.
I would like to craft a table.
I would like to eat food.
I would like to drink water.
I would like to rest.
I would like to build stone tools to defend myself against monsters.
I would like to build bridges to cross lakes.
I would like to dig tunnels to hide from monsters.
I would like to block arrows with stones.
I would like to dig through walls to surprise skeletons.
I would like to seek shelter in caves.
I would like to build plantations of saplings and defend them against monsters.
I would like to eat the growing fruits to ensure a steady food supply.
I would like to place a table.
I would like to eat a cow.
I would like to place a plant.
I would like to defeat a zombie.
I would like to place stone.
I would like to eat a plant.
I would like to defeat a skeleton.
I would like to wake up.
I would like to place a furnace.

Write all game objectives numbered list. For each objective, list its requirements.
1. Collect Wood: No requirements
2. Place Table: Requires Collect Wood
3. Eat Cow: No requirements
4. Collect Sampling: No requirements
5. Collect Drink: No requirements
6. Make Wood Pickaxe: Requires Place Table
7. Make Wood Sword: Requires Place Table
8. Place Plant: Requires Collect Sampling
9. Defeat Zombie: No requirements
10. Collect Stone: Requires Make Wood Pickaxe
11. Place Stone: Requires Collect Stone
12. Eat Plant: Requires Place Plant
13. Defeat Skeleton: No requirements
14. Make Stone Pickaxe: Requires Collect Stone
15. Make Stone Sword: Requires Collect Stone
16. Wake Up: No requirements
17. Place Furnace: Requires Collect Stone
18. Collect Coal: Requires Make Wood Pickaxe
19. Collect Iron: Requires Make Stone Pickaxe
20. Make Iron Pickaxe: Requires Place Furnace, Collect Coal, and Collect Iron
21. Make Iron Sword: Requires Place Furnace, Collect Coal, and Collect Iron
22. Collect Diamond: Requires Make Iron Pickaxe

Write all actions as a numbered list. For each action, list its requirements.
1. Move West: Flat ground west of the agent.
2. Move East: Flat ground east of the agent.
3. Move North: Flat ground north of the agent.
4. Move South: Flat ground south of the agent.
5. Do: Facing creature or material; have necessary tool.
6. Sleep: Energy level is below maximum.
7. Place Stone: Stone in inventory.
8. Place Table: Wood in inventory.
9. Place Furnace: Stone in inventory.
10. Place Plant: Sapling in inventory.
11. Make Wood Pickaxe: Nearby table; wood in inventory.
12. Make Stone Pickaxe: Nearby table; wood, stone in inventory.
13. Make Iron Pickaxe: Nearby table, furnace; wood, coal, iron an inventory.
14. Make Wood Sword: Nearby table; wood in inventory.

```
15. Make Stone Sword: Nearby table; wood, stone in inventory.
16. Make Iron Sword: Nearby table, furnace; wood, coal, iron in inventory.
17. Noop: Always applicable.

Observation Example:
You took action move_west.

You see:
- water 5 steps to your south-west
- grass 1 steps to your west
- sand 4 steps to your south-west

You face grass at your front.

Your status:
- health: 9/9
- food: 9/9
- drink: 9/9
- energy: 9/9

You have nothing in your inventory.
```

### C.6 MINECRAFT

**Example input**

```
Instruction Manual:
You are in Minecraft and your goal is to find a forest biome. You are not allowed to craft anything.

In your observation, you are provided the amount of space an object takes in your field of view.
Note that objects of the same size takes more space when they are closer to you.

Observation Example:
You took action Move East.

Coordinate (-253.12,71.75,248.64). Facing east.

You're not aiming at any block.
Around you:
 - leaves, 3.96 blocks away, above you to north-west
 - wood, 4.24 blocks away, to north-west
 - grass block, 1.34 blocks away, below you to north-east
 - dirt, 3.33 blocks away, below you to north-east
 - stone, 4.67 blocks away, below you to north-east

You see:
 - grass block, 1.36 blocks away, below you to north-west, taking 51% of screen
 - sand, 8.29 blocks away, below you to south-west, taking 4% of screen
 - leaves, 4.47 blocks away, above you to north-west, taking 17% of screen
 - grass, 5.49 blocks away, above you to north-west, taking 1% of screen
 - wood, 11.07 blocks away, above you to north-west, taking 0% of screen
```

## D   ADDITIONAL EXPERIMENTAL RESULTS

### D.1   HUMAN BASELINE

3 players (including the authors) who are very familiar with the environments and API played the games through the SmartPlay interface. Each human player performed **3 rounds** of Bandit, RPS; **1 round** of Hanoi, Crafter, Minecraft; **5 rounds** of MessengerL1, MessengerL2. We report the final average score over all trials and all players.

### D.2   NORMALIZED HUMAN SCORE

Given the game score of an LLM on game $g$, $s_g^{(\text{raw})}$, we compute normalized human score $s_g$ from the human baseline on $g$, $s_g^{(\text{human})}$, and the minimum possible game score $s_g^{(\text{min})}$:

$$s_g = \frac{s_g^{(\text{human})} - s_g^{(\text{raw})}}{s_g^{(\text{human})} - s_g^{(\text{min})}}$$

### D.3   RAW SCORES

| LLM | Bandit | RPS | Hanoi | MessengerL1 | MessengerL2 | Crafter | Minecraft |
|---|---|---|---|---|---|---|---|
| Human Baseline | 45 | 43 | 3 | 1 | 1 | 2680 | 1 |
| GPT-4-0613 | 45.09 | 39.25 | 2.5 | 0.8 | 0.85 | 700 | 0.61 |
| GPT-4-0314 | 43.86 | 42.05 | 2.7 | 0.74 | 0.93 | 845.6 | 0.592 |
| text-davinci-003 | 46.92 | 17.0 | 1.5 | 0.24 | -0.07 | 186.25 | 0.449 |
| Claude | 32.43 | 20.3 | 2 | -0.12 | 0.2 | 143.3 | 0.5 |
| Bard | 38.85 | 12.9 | 2 | 0.22 | -0.21 | 112.3 | 0.54 |
| llama-2-13b | 22.33 | 15.05 | 1.1 | -0.76 | -0.745 | 115.3 | 0.606 |
| llama-13b | 30.5 | 21.4 | 1 | -0.68 | -0.885 | 100.2 | 0.5 |
| vicuna-13b | 28.81 | 7.1 | 0.2 | -1 | -0.76 | 56.7 | 0.43 |

Table 4: Comparison of performance of different LLMs in terms of average score on BanditTwoArmedHighLowFixed-v0, RockPaperScissorBasic-v0, Hanoi3Disk-v0, MessengerL1-v0, MessengerL2-v0, Crafter-v0, MinedojoCreative0-v0.

