# OpenReview forum: "SmartPlay : A Benchmark for LLMs as Intelligent Agents"
_ICLR.cc/2024/Conference — ICLR 2024 poster_

### Official Review · Reviewer_itPd · 2023-10-26

**Soundness:** 3 good
**Presentation:** 3 good
**Contribution:** 4 excellent
**Rating:** 8
**Confidence:** 4

**Summary:**

The paper presents SmartPlay, a benchmark that turns a number of control-agent environments applicable for LLMs by providing observations and actions in text format. The benchmark involves 20 unique evaluation settings, together spanning 9 different important capabilities highlighted by the paper. Results with different LLMs indicate that GPT-4 and its variants outperform other LLMs across the different tasks.

**Strengths:**

This type of work, presenting benchmarks and evaluation settings, is direly required in the LLM-as-agents space. Recent publications have each come up with their own ways of interfacing with environments and defining new tasks, and it is impossible to tell which agents are "more capable" than others. This work is a clear stepping-stone in defining standardized sets across different domains.

### Originality

Very original, no other exhaustive benchmarks in this domain exist.

### Quality

Moderate to high quality. The benchmark is exhaustive in terms of different environments and challanges included, as well as conducting basic baselines. However some details and potentially expected benchmarks are missing (see weaknesses).

### Clarity

Paper is well structured and easy to read for the most part.

### Significance

This paper, and the benchmark, can be important motivators and stepping stones for people to create comparable results between different LLM agent solutions. While I do not expect this to become de-facto way of measuring agent performance, I believe it will serve as a fixed benchmark, and in future people can improve upon it when needed. I believe this is a very significant contribution for ICLR readers.

**Weaknesses:**

The paper omits bunch of important details and is vague at parts, which is why I am setting my recommendation to "borderline accept" in the first rating.

### Missing details

Some important details to function as a benchmark are missing, such as  _how_ the evaluation of new LLMs/agents should be done (e.g., number of episodes, at minimum. Are users allowed to change the environment? Can users change the prompt manually before feeding to the agent?)

Details/motivation of the different capabilities is missing (see questions). I feel this need to be clarified and carefully motivated for a solid case why these capabilities matter and how they are properly measured.

Other important details (e.g., human baseline collection) are missing. See questions.

### (Minor) Missing baselines/experiments

While having a fixed, single way of defining prompts makes the benchmark fixed and results comparable across models, I think there should be an option to try different techniques of prompting the models. E.g., chain-of-thought [1] can, with simple modifications to the prompt, improve results in LLM space (see questions).

**Questions:**

1) How were numbers for the spider plots in Figure 2 came up with? On left it shows three degrees for every capability, but it is not obvious if it directly maps to different levels in the spider plot. If numbers on spider plot should reflect these three levels, I'd recommend adding the labels to the spider plots so reader knows to connect the two. Additionally Table 3 in the Appendix has four degrees for some of the capabilities (e.g., long text understanding), but this is not shown in Figure 2. Is this to save space or an oversight? If anyway possible, I'd recommend keeping the two consistent to avoid confusion.
2) How were the capabilities chosen, and how would you exactly define their different degrees and their importance to the LLM research? By reading the paper, it seems they stem from the environments you chose, but they still come out as somewhat vague (e.g., what is "Randomness", exactly? Randomness in what? What is "Reasoning with object dependencies"? What is "rollback" error handling?). A valuable contribution would be to add as exact definitions as you can come up with, and detail them in a new section of the paper. The choice of your games helps you keep grounded to what can be measured, and by defining the metrics accuratelly, other researchers can bring new environments / dimensions into the mix.
3) Likewise, can you provide exact details on the environments used in your benchmark package? While the code will allow exact reproduction, it might get updated as it lives on Github, so having a one solid description of the originally proposed setup would be beneficial.
4) Have you experiment with ways to improve model performance by, e.g., chain-of-thought [1] prompting? These seem to improve models' performance across different tasks, and I feel this should be supported by this benchmark. E.g., you could add a flag or mutator or wrapper to your environment that adds/modifies the instructions/observations. Seeing initial results with these would strengthen the paper, but not necessary; my core request is to think these possibilities and also support them.
5) How was the "human baseline" obtained, exactly? Did they play the games through the same exact API as the LLMs play (i.e., via text)? Were they familiar with the environments beforehand? Were there multiple human players? Likewise, how was the zero value defined in Table 2? Was it a random agent or literal 0 value for the score?

References:
- [1] Jason Wei, *et al*. Chain-of-thought prompting elicits reasoning in large language models. In NeurIPS, 2022.

--------------------------------------------------------
## 16th Nov

I read and acknowledged authors' rebuttal, and have increased my score from 6 to 8.

---

> ### Author Response · Authors · 2023-11-11
>
> Thank you for recognizing the originality, quality, clarity, and significance of our work.
>
>
> Here are our responses to your questions and concerns:
>
>
> W1.1 (evaluation protocols) Please see Section 3.1 Environment Interface and Evaluation Protocol, and Table 1 for all the details required, including num episodes for each game, and what the user may or may not change.
>
>
> W1.2 (Motivation for different capabilities) Please see Q2
>
>
> W1.3 (Human baseline) Please see Q5.1
>
>
> W1.4 (Different prompts) To be fair to all LLMs, the experiments we conduct (page 7, Section 4) uses the popular minimalist prompt of “let’s think step by step”. However, as explained in the general rebuttal (The SmartPlay Interface section), SmartPlay is compatible with any prompting frameworks. We hope that SmartPlay will help future research on creating
>
>
> Q1 (Figure 2 - Table 3) Apologies for the potential misunderstanding, please note that only two capabilities, “text understanding” and “Learning from Interactions” have 4 levels in both Table 3 and Figure 2, we did not notice any disagreements.
> Thanks for the suggestions on adding the numbers (from Table 3) to spider plots, we will update the figures with the numbers for the final version of our paper.
>
>
> Q2 (Motivation for different capabilities) Identifying the 9 capabilities is part of the contribution of the paper. Our choice and definitions are largely motivated from challenges that game-design try to capture [1]. "Randomness" refers to understanding of randomness, which is important because many real-world events are probabilistic (covered in introduction).  "Rollback" is discussed in tower-of-Hanoi Section 2.4 as undo incorrect moves.
>
> We discussed in detail how each challenge instantiates in individual games in Section 2, but we did not offer a formal introduction.
> Thank you for pointing out this omission, we will update the final version of our paper with a formal introduction to how each level of the 9 capabilities are defined.
>
>
> Q3 (Environment Details) We cover environment rules, mechanics, and observation in Section 2 “Games in SmartPlay”. In addition, we provide example observations from all environments in the appendix.
>
>
> Q4 (Chain-of-thought) To be fair to all LLMs, the experiments we conduct (page 7, Section 4) uses the popular minimalist prompt of “let’s think step by step”, which is a form of chain-of-thought prompting. It is unclear how the specific ``chain-of-thought’’ prompting paper cited in the review should be applied to our setting, since the paper depends on few-shot examples of similar reasoning, but the 6 games in SmartPlay differ from each other greatly in terms of game mechanics. In addition, we explicitly avoid heavily engineered prompting frameworks in our experiments like ReAct or Voyager since results with those frameworks could soon be out-of-date given the current speed of the academia.
>
>
> Q5.1 (human baseline) 3 players (including the authors) who are very familiar with the environments and API played the games through the API. We will add clarification to this in the paper.
>
>
> Q5.2 (values in table 2). Values in table 2 are normalized according to human baseline with (score-min_score)/(human-min_score). A score of 0 corresponds to achieving the minimum possible score in the game (failing all tasks and performing worse than random). We will add clarification to this in the paper.
>
>
> [1] Raph Koster. Theory of fun for game design. ” O’Reilly Media, Inc.”, 2013.

---

> > ### Comment · Reviewer_itPd · 2023-11-13
> >
> > Hey, thank you for your prompt replies!
> >
> > **Summary**: I thank authors for addressing my questions. I am leaning towards increasing my score. However, since I have not seen the exact promised changes on the manuscript nor I have not seen the source code of the environment (which is very core part of the manuscript), I can not with good consciousness increase my score at this point (13th Nov). As mentioned in text below, I have seen too many examples of authors of a manuscript promising changes or open sourcing the code, but never delivering it. I apologize for what this may seem ("holding hostages"), but in a paper that is specifically about an open benchmark, not being able to view the code is a severe limitation at the review phase.
> >
> >
> > **Comment on updating manuscript**: While I appreciate the replies to all of my questions, I'd also recommend uploading an updated PDF that has the mentioned changes (where-ever reasonable) in them. This will help reviewers to gain confidence in your work and unambigiously show how you changed the manuscript (I'd even recommend highlighting changed areas in the rebuttal version). I am bringing this up as I have been bitten one too many times by authors of other manuscripts claiming that they will update the manuscript, but never did so even when the work was accepted.
> >
> > Likewise, many times authors in other manuscripts claimed "they will share the open source code", but never did so. I understand some organizations and companies might have harsh limits on this, especially for review phase, but I'd appreciate if reviewers had a chance of at least confirming the code exists and contains what is promised. This is especially important for a benchmark paper like this, as the code is the core contribution to the community.
> >
> >
> > **W1.1**: Ah, I was obviously blind. Thanks for pointing out that this information was in Section 3.1! The only thing I feel needs bit of clarification here are unambigiously defining "history" and "rollout length". I assume these are "history length in number of environment steps" and "rollout length in number of agent queries"? I would appreciate defining the unit of these variables.
> >
> > **W1.4**: Thank you for the reply and clarification. Indeed I agree that the benchmark is modular and allows such modifications in future (and a proper evaluation of such changes, given the otherwise fixed benchmark).
> >
> > **Q1**: (Figure 2 - Table 3): Thanks for the clarification! I'd highlight this difference between different capabilities having different number of rankings to avoid further confusion. Also see my generic comment at the top.
> >
> > **Q2**: Thank you for pointing out the clarifications in Section 2, but as you noted, I agree that a more formal definition of capabilities in a different section would be more readable.
> >
> > **Q3**: Thanks for pointing out Section 2 again. On a second thought, defining the benchmark in the paper would be very cumbersome effort (if not possible). However this highlights the need for code access for reviewers, as it is, in essence, defining big chunk of the work.
> >
> > **Q4**: Thank you for the clarification. This also connects to the weakness W1.4, which you addressed by also bringing up the modularity of the framework. I agree that this benchmark is better set in as vanilla setup as possible, while others can take it and build on it with more heavier frameworks (if they wish to experiment with them).
> >
> > **Q5.1**: Thanks for the clarification. For baseline purposes this is appropriate, but please do clarify this in detail in the paper, as it is an important detail to know if humans were unfamiliar or familiar with the interface and games.
> >
> > **Q5.2**: Thanks for the clarification.

---

> > > ### Author Response · Authors · 2023-11-14
> > >
> > > Dear Reviewer itPd,
> > >
> > >    Thank you for the detailed improvement suggestions and the insightful suggestions, especially for pointing out the necessity for a formal definition of LLM agents' capabilities.
> > >
> > >    We have updated the manuscript (See general rebuttal "Edits to manuscripts" for the change details) to address your concerns and improvement suggestions. We are still working on adding ticks to the figures since some axis (capability) in the spider chart use a different scale, and our plotting function currently lacks the support for such functionality. We will continue to work on this minor problem.
> > >
> > >    In the meantime, please let us know if we are able to address all your remaining concerns.

---

> > > > ### Comment · Reviewer_itPd · 2023-11-16
> > > >
> > > > Dear authors,
> > > >
> > > > Thank you for your prompt reply! I acknowledge you have taken my comments into account. I also thank you for open sourcing the code for review, and I am happy to increase my review score knowing the state of the code. I have increased my score from 6 to 8 to signal my clear acceptance vote.
> > > >
> > > > Some final comments I'd like to see in future revisions (e.g., camera ready:
> > > >
> > > > - The Section 2, while containing the information it should have, should be rewritten for clarity and with more examples. I highly recommend you pass the text through a colleague who has not written the paper to ensure the writing is clear enough for a new reader.
> > > > - Since the code will be a major contribution of this work, I highly recommend some generic steps to ensure the code is approachable by others, and will live on to serve as a solid benchmark and a point to build on:
> > > >   - Apply some common code formatting rules, e.g., Black, to ensure the style is consistent across files. This greatly improves the readibility
> > > >   - Offer concise (minimal) example on how to run the environment, e.g. ideally few 10s of lines. The experiment.py provided is a complete example but very verbose, and it is hard to tease out what is your environment.
> > > >   - Add some documentation to environments for more details how they work and how they are defined. Also add complete installation/usage description to the README (the current one is decent as it is)
> > > >   - (Optional, minor) if you have bandwidth, also sharing the recordings from your experiments running different LLMs would allow people to already try out things without running LLMs. Not super necessary, but a thought, since running LLMs can get expensive.

---

> > > > > ### Author Response · Authors · 2023-11-17
> > > > >
> > > > > Dear Reviewer itPd,
> > > > >
> > > > >    Thank you for all detailed and insightful advice and comments on the code-base and readability. We care deeply about improving the ease-of-use of the benchmark, and improving the code-base will be our top priority.

---

### Official Review · Reviewer_j3e9 · 2023-10-30

**Soundness:** 2 fair
**Presentation:** 2 fair
**Contribution:** 2 fair
**Rating:** 5
**Confidence:** 4

**Summary:**

Summary:

The paper introduces "SmartPlay," a benchmark designed to evaluate Large Language Models (LLMs) in the role of intelligent agents. The authors emphasize the growing potential of LLMs in intelligent agents and next-generation automation, highlighting the need for a systematic benchmark to assess their capabilities.

Main Contributions are as follows:
1. **Introduction of SmartPlay**: The paper presents SmartPlay as both a challenging benchmark and a methodology for evaluating LLMs as intelligent agents.
2. **Diverse Game Set**: SmartPlay includes a set of 6 different games, such as Rock-Paper-Scissors, Tower of Hanoi, and Minecraft, each with its unique setting. These games collectively offer up to 20 evaluation settings and infinite environment variations.
3. **Capability Assessment**: Each game in SmartPlay is designed to challenge a subset of 9 crucial capabilities of an intelligent LLM agent. These capabilities include reasoning with object dependencies, planning ahead, spatial reasoning, learning from history, and understanding randomness.
4. **Comprehensive Evaluation**: Through the diverse set of games and challenges, SmartPlay aims to provide a comprehensive evaluation of LLMs’ abilities as agents.

**Strengths:**

1. **Definition and Problem Formulation**: The paper introduces a new benchmark, SmartPlay, specifically designed to evaluate Large Language Models (LLMs) as intelligent agents. This represents a novel approach to addressing the current gap in systematic evaluation methods for LLMs in agent-based roles.
2. **Well-Defined Benchmark**: SmartPlay is presented in a structured manner, with explanations of the different games and the specific capabilities they assess. This clarity aids in understanding the paper's objectives and the proposed methodology.

**Weaknesses:**

1. **concerns on fairness for different LLM**: The author solely relied on a pre-trained LLM for evaluation in these environments. However, since various language models are trained on different corpora and some remain undisclosed (such as close-source OpenAI language models), it becomes challenging to ascertain whether the LLM has been exposed to these specific environments. For instance, GPT possesses extensive knowledge of Minecraft, whereas LLAMA has relatively limited knowledge in this area. Consequently, ensuring fairness in direct comparisons of evaluations becomes difficult. To address this issue, I recommend that the author gather sufficient data for each environment and evaluate the capabilities of different language models separately using zero-shot learning, few-shot in-context learning, and instruction tuning approaches.
2. **concerns on 3d spatial reasoning performance of LLM**: I noticed that current Large Language Models (LLMs) struggle with 3D spatial reasoning in environments like Minecraft. This could be due to the lack of visual information in their training data, making it difficult to directly apply LLMs to tasks in this dataset. The author attempted to describe visual images using text, but previous experiments have shown this approach to be impractical. Additionally, the source of these visual descriptions was not explained by the author. Can models like LLaVA, GPT-4V, and Flamingo overcome these challenges by being fine-tuned with visual images and other relevant information?
3. **concerns on difficulty computation**: The setting of difficulty for each game seems quite arbitrary. How are these difficulties determined? Are they based on human evaluation? I suggest the author provide clear explanations regarding this.
4. **concerns on prompt design for different llms**: Despite instruction tuning, the performance of LLM still relies on prompt design. It is best to conduct explicit experiments and explanations on prompt design.

**Questions:**

1. The author can use **one** simulator to assess the various abilities of the LLM agent, rather than relying on multiple simulators for each ability. For instance, Minecraft offers an open world with numerous engaging tasks that require long-term planning and learning from interactions. However, the author only utilized Minecraft to evaluate the LLM agent's 3D spatial reasoning skills. Another convenient and popular option is a readily installable simulator package that is user-friendly.
2. **Comparison with Existing Benchmarks**: The paper could be strengthened by comparing SmartPlay’s performance and effectiveness in evaluating LLMs against existing benchmarks or evaluation methodologies, if any are available.
3. There are some typos and table type errors.

**Details Of Ethics Concerns:**

Null

---

> ### Author Response · Authors · 2023-11-11
>
> Thank you for recognizing our definition of specific capabilities for an LLM agent.
>
> Here are our responses to your questions and concerns:
>
>
> W1 (fairness due to LLM knowledge difference): Thank you for pointing out this important concern. Test set contamination is indeed one of the key challenges of LLM evaluation as noted by [1]. SmartPlay attempts to mitigate contamination by 1) featuring 6 different games 2) featuring games (Messenger and Crafter) that are created for research with little online documentation 3) Using games with procedural generation (Messenger, Crafter, Minecraft).
>
> In addition, games seem to be quite robust to memoization due to their nature of exponential branching states, (in section 4.2) as we experimentally observe with the Tower of Hanoi. Although all LLMs could provide a memoized solution at the starting configuration where all disks are on the first rod (some may even write out the recurrence for the solution), most LLMs could not solve the problem and get confused quickly after a few moves, especially when the disks are distributed over all three rods.
>
>
> W2 (VLMs): Please note that playing games requires accurate object location/interaction perception. The spatial reasoning aspect can go beyond object recognition. Controlling in-game agents requires precise and accurate identification of interactions and changes in the environment [2], current open-source VLMs lack such capabilities. We are hopeful for future VLMs will do better at spatial reasoning aspects than LLMs potentially, but the goal of our work is to show that LLMs are generally weaker at spatial reasoning.
>
>
> W3 (Difficulty definition): Please see Figure 2 (left table) or Table 3 (Appendix page 14) for how each game challenges different capabilities, and how we define the difficulty levels. For continuous difficulty spectrums like text understanding, reasoning, instruction following, interactions, spatial reasoning, planning, we define the levels by defining difficulty bins. For categorical difficulty spectrums like generalization, randomness, error-handling, we define the levels by enumerating all options. Thank you pointing out this omission, we will include more information on how we defined the difficulty levels in the final paper.
>
>
> W4 (Prompt design for LLMs): As explained in the general rebuttal (The SmartPlay Interface section), SmartPlay is compatible with any prompting frameworks.
>
> To be fair to all LLMs, the experiments we conduct (page 7, Section 4) uses the popular minimalist prompt of “let’s think step by step”.
> Although some LLMs might do better at specific types of prompts, such behavior is generally considered undesirable for downstream applications and should not be explicitly accommodated.
>
> New prompting frameworks are invented constantly, and future LLMs may be tuned accordingly. If we include complex prompting frameworks, we may need to constantly update the core evaluation frameworks, making past results incompatible.
>
>
> Q1 (single simulator like Minecraft): In regards to user-friendliness, SmartPlay widely supported OpenAI gym protocol, and could be installed easily with pip.
>
> In regards to only using one simulator like Minecraft, we note that this will significantly reduce the diversity of the benchmark and increase the risk of test set contamination. In addition, ALL existing LLM models fail on simple navigation tasks in Minecraft, not to mention harder tasks that involve multiple dependencies (i.e., mine diamond), and may depend on navigation. Therefore, it is unclear if a single simulator could capture the challenges at different difficulty levels.
>
>
> Q2 (Existing benchmarks): Apologies for the potential misunderstanding, we are unclear about how the "performance and effectiveness in evaluating LLMs" could be defined and compared. SmartPlay is created to benchmark specific capabilities of *LLMs as Agents* and is currently one of its kind as noted by reviewers itPd, Jri2. Therefore, there is no similar benchmark for direct meaningful comparison.
>
>
> Q3 (Typos) Thank you for pointing out this important issue. We will proof-read the paper for spelling and correctness.
>
>
> [1] Llama 2: Open Foundation and Fine-Tuned Chat Models
>
> [2] SPRING: GPT-4 Out-performs RL Algorithms by Studying Papers and Reasoning

---

> > ### Comment · Reviewer_itPd · 2023-11-13
> >
> > Hey, replying to Q1 ("using single simulator like Minecraft")
> >
> > While I agree Minecraft is open-ended and offers possibility to create numerous of different testing scenarios, I agree it still is one environment but I'd even emphasize the practicality of it. This benchmark is meant to live on longer as a fixed set of tasks so people can easily compare their LLM solutions. Minecraft is comparatively difficult to install (yes, it can be one `pip install` command but often has headaches), and is slow to run. It also relies on numerous of dependencies (python packages, remote repositories, etc), which may break in future. As such, I'd highly encourage using as dependency-less code as possible for benchmark suite like this.
> >
> > However, Minecraft will still serve as its own suite of complicated tasks and milestones (e.g,. beating Ender dragon). I feel this work is more like an "unit test" (akin to [1]) / simple benchmark for LLM agents, where you can iterate faster, while Minecraft is then one of the areas where you can scale up / demonstrate the utility of the LLM agents.
> >
> > Regarding the "LLMs fail in simple navigation in Minecraft": some related works have shown the contrary [2, 3]. **However**, this is the exact reason why I think benchmarks like this are crucial right now: these works define the way agent interacts with the environment in different ways (or, indeed, define "navigation" in different ways). While the "right way" of defining the interface between an LLM agent and environment is still up for a debate, having _something_ fixed (like this benchmark) would remove one point of variation between different works.
> >
> > References:
> > - [1] Osband, Ian, et al. "Behaviour suite for reinforcement learning." arXiv preprint arXiv:1908.03568 (2019).
> > - [2] Wang, Guanzhi, et al. "Voyager: An open-ended embodied agent with large language models." arXiv preprint arXiv:2305.16291 (2023).
> > - [3] Wang, Zihao, et al. "Describe, explain, plan and select: Interactive planning with large language models enables open-world multi-task agents." NeurIPS 2023 (https://arxiv.org/abs/2302.01560)

---

> > > ### Author Response · Authors · 2023-11-13
> > >
> > > Dear reviewer itPd,
> > >
> > >    Thanks for engaging in the active discussion. We are fully on board with your understanding of this work as a "unit test".
> > >
> > >    On the other hand, please note that both works you cited for navigation Voyager [1], DEPS [2] actually rely heavily on manually-programmed code-bases for navigation (Voyager), or trained policy (DEPS). In fact, Voyager and DEPS use LLMs as more like a coding or tool-calling planner. The LLMs do not take in 2D/3D observation, and are not able to react frequently to objects/entities in the environment (all reaction/manipulation is done by the low-level learned/hand-coded primitives).

---

> > ### Comment · Reviewer_j3e9 · 2023-11-17
> >
> > Thank you for the author's reply.
> > 1. The language navigation task conducted in Minecraft to examine spatial reasoning seems very strange. Shouldn't this be a visual navigation task? I see the author's description of the visual image, but it does not accurately describe the current scene structure, which is crucial for navigation tasks.
> > 2. Q2: Compared to AgentBench [AgentBench: Evaluating LLMs as Agents]

---

> > > ### Author Response · Authors · 2023-11-17
> > >
> > > Dear Reviewer j3e9,
> > >
> > > Thank you for engaging in the discussion. Below are answers to your concerns:
> > >
> > > 1. Indeed, the Minecraft task in SmartPlay visual navigation task. The agent needs to traverse different terrains and biomes in order to find "plains". Spatial reasoning is instantiated in Minecraft since the agent needs to identify terrain type based on "above, blow, left, right", and if it's blocked based on distance.
> > >
> > > In regards to the "scene structure" mentioned by the follow-up question, do you mean arrangement and organization of objects?
> > >
> > > Note that the "You see: xxx" is a direct translation of the segmentation map of the visual observation, augmented with depth information. Therefore, the arrangement of objects is preserved enough to find very coarse biomes like the "plains", as verified by the human players who were able to succeed in the games relying only on the API. (We have added section B in the appendix to explain how the description is generated)
> > >
> > >
> > >
> > >
> > > 2. First, please note that AgentBench https://openreview.net/forum?id=zAdUB0aCTQ is a concurrent work. Secondly, please note that the two benchmarks study different aspects. SmartPlay focuses on providing insights into **different capabilities** of intelligent LLM agents using "plain text", and AgentBench is more API focused with most challenges requiring the LLM to have some coding abilities. In addition, SmartPlay provides human baseline scores for benchmark tasks, which sets future goals for LLM agents.
> > >
> > > Benchmarking LLMs as intelligent agents is an important TODO for the community. We are genuinely happy about the attempts in the community to tackle this challenge. However, it is important to note that the concurrent work (AgentBench) serves a different purpose with a program-based focus, whereas SmartPlay focuses on providing insights using a curated set of simple games.

---

### Official Review · Reviewer_E24b · 2023-10-31

**Soundness:** 3 good
**Presentation:** 2 fair
**Contribution:** 2 fair
**Rating:** 6
**Confidence:** 2

**Summary:**

This paper aims to address the gap in systematic benchmarks for evaluating Large Language Models (LLMs) in the context of intelligent agents. The authors propose SmartPlay, a benchmark consisting of six diverse games, each designed to test different capabilities vital for intelligent agents, such as reasoning with object dependencies, planning, and spatial reasoning. The games include Rock-Paper-Scissors, Tower of Hanoi, and Minecraft, among others. They claimed both benchmark and methodology contribution of testing LLM performance beyond language-solely-based tasks. Also, they tested some well-known LLM on the proposed game benchmarks, including GPT variants, llama variants, and etc.

---
# Post Rebuttal

I appreciate the efforts made by the authors. Their rebuttal clarify lots of my concerns, and thus, I raised my scores. However, I believe I am not an expert in the field of LLM-Agent --> I am just okay with its acceptance.

**Strengths:**

This paper takes a step towards a crucial need in the field of LLM for a standardized benchmark to evaluate the agent-like abilities of LLMs. SmartPlay is presented as both a benchmark and a methodological tool, which may be a good contribution to the research community. By offering a variety of games that test a comprehensive set of agent capabilities, the benchmark allows for a detailed assessment of LLMs beyond language-solely-based tasks. The commitment to providing an open-source benchmark (referenced GitHub repository) is commendable and encourages community engagement and continuous improvement. The paper is well-structured and provides good explanations of the games and the intended capabilities they aim to evaluate.

Extensive evaluation of existing LLM are provided in Table2. Some analysis are also included to emphaize the need of the proposed benchmark. Also, the results show the gap between open-sourced LLM and commercial LLM.

**Weaknesses:**

I believe the starting point, i.e. evaluating the LLM in the context of intelligent agent, is crucial to our community. However, I am unsure if the proposed games can well evaluate this aspects: (1) the games are still too simple to solve real-world challenges; (2) It is unclear which games correspond to which nine abilities to which levels; (3) why human has all 1 for all the games? why davinci-model has 1.04 on bandit?

Besides, there may be concerns regarding the scalability of the benchmark and whether it can keep pace with the rapid advancements in LLMs.

Additionally, the paper might benefit from a more detailed discussion on the implications of these benchmarks in real-world scenarios and how they reflect the complexities of actual agent tasks.

**Questions:**

It seems the github repo is empty? https://github.com/LLMsmartplay/SmartPlay

Please also address the point mentioned above.

---

> ### Author Response · Authors · 2023-11-11
>
> Thank you for acknowledging that SmartPlay takes a step to satisfy a critical need, and the benchmark provides a wide range of games.
>
> Here are our responses to your questions and concerns:
>
> W1.1 (Games too simple) Thank you for pointing out this concern. Do you mind specifying what kind of challenge in SmartPlay is too simple for LLM agents?
> Games in SmartPlay capture multiple aspects of fundamental challenges for LLM agents: text understanding, reasoning with dependencies, instruction following, planning, generalization, understanding and estimating randomness, error/mistake handling, and spatial reasoning. Each challenge is further classified into multiple degrees according to Figure 2 and Table 3.
>
> In addition, the game rules and mechanics are intuitive to understand by design, so that they pose fair challenges to all LLMs. Our research goal for SmartPlay is not to be the hardest benchmark, but to set a goal that is reachable in a short time-frame yet formidable to require new breakthroughs. We note that all state-of-the LLMs greatly under-perform human players in SmartPlay, and there is a clear separation in performance between different LLMs.
>
> Finally, we note the potential of extending SmartPlay with more complicated games as described in the general rebuttal (The SmartPlay Interface section).
>
> W1.2 (Game - Ability correspondence) Please see Table 3 (Appendix page 14) for how each game challenges different capabilities.
>
> W1.3 (human normalized scoring for Table 2) We normalize the LLM scores by the ratio with human expert score on all games. Therefore, human baseline score is always 1, and text-davinci achieves slightly better score than human expert on bandit, so has a score of 1.04. For un-normalized scores, see Table 4 (Appendix page 18).
>
> W2 (scalability) we note that it is easy to extend SmartPlay with more complicated games as described in the general rebuttal (The SmartPlay Interface section).
>
> W3 (real-world agent tasks) Please note that the field of LLMs as agents has just started to grow. With OpenAI releasing new GPTs models earlier this week, it is unclear how one would predict what type of tasks would eventually be "real-world". Although there has been growing attention on web-browsing, gaming, and VR/AR applications, we note that none of these agent applications are at the scale of real-world agent tasks.
>
> Therefore, we take the perspective of identifying fundamental challenges with agents: text understanding, reasoning with dependencies, instruction following, planning, generalization, understanding and estimating randomness, error/mistake handling, and spatial reasoning. We believe those challenges not only exist in the field of games, but also ALL real-world agent tasks.
>
> Q1 (Empty GitHub Repo): Our repo is ready but cannot be anonymized due to legal issues. We guarantee public release of (unanonymized) code immediately upon acceptance.

---

### Official Review · Reviewer_Jri2 · 2023-11-04

**Soundness:** 3 good
**Presentation:** 3 good
**Contribution:** 3 good
**Rating:** 8
**Confidence:** 3

**Summary:**

The paper introduces SmartPlay, a benchmark with 6 games, designed to evaluate the capabilities of recent large language models (LLMs) when applied as agents in intelligent automation. The authors identify 4 key challenges important for general intelligent LLM agents but not captured in previous benchmarks: 1) long-horizon planning and execution, 2) understand the odds, 3) spatial reasoning, 4) learn from interactions or mistakes.  The authors claim that each of the games of the new benchmark offers a unique challenge mentioned previously.  Overall, SmartPlay is positioned to push the boundaries of current LLM evaluation methodologies. By systematically assessing the performance of LLMs as agents across a range of games and challenges, SmartPlay aims to provide insights into the current state of LLMs and identify gaps that need to be addressed for the advancement of intelligent agents.

**Strengths:**

1, SmartPlay is a good benchmark for evaluating the performance of large language models (LLMs) as agents. It introduces a diverse range of games carefully chosen to assess different critical capabilities required for intelligent agents, making SmartPlay a well-structured and challenging platform.

2, The paper is well-written. The authors clearly articulate the need for such a benchmark and provide enough background and related works to well-position the benchmark.

3, In addition to introducing the benchmark, the authors also conduct a comparative analysis of current state-of-the-art LLMs using SmartPlay. This comparison is crucial as it validates the effectiveness and rigor of the benchmark, and it provides a snapshot of the current landscape of LLMs' abilities as agents. The findings from this analysis enhance the understanding of LLMs' strengths and weaknesses, identifying areas that require further development and offering clear directions for future research.

**Weaknesses:**

1, Evaluation metrics proposed in the paper are commonly used in the domain of reinforcement learning (RL). While these metrics are established and provide a common ground for comparison, they may not be entirely suited to capture the unique nuances of planning and reasoning that are specific to LLMs functioning as agents. The paper could be strengthened by proposing or developing novel metrics that are tailored to the particular dynamics of LLMs' operational framework, offering a more precise measurement of their planning, reasoning, and adaptability skills in agent-based scenarios.

2, Is it possible to craft the prompt to show the flow of reasoning and planning when LLMs as agents play the games?

**Questions:**

See weaknesses section

---

> ### Author Response · Authors · 2023-11-11
>
> Thank you for the recognition that our proposed benchmark is well-structured and challenging, and our analysis is comprehensive.
>
> Here are our responses to your questions and concerns:
>
> W1 - Thank you for pointing out this important aspect of LLM evaluation. Indeed, the original reward functions designed for RL algorithms are not the best representations of agent achievements in many games. Therefore, we introduced "game score" (bottom of page 6, section 3.3), which is defined to best represent the accuracy of choices or the progress toward task completion. The normalized game score in table 2 further denotes the gap between human LLM capability and human capability at specific games.
> In regards to capability-specific metrics, we believe deeply in the benefit of having capability-specific metrics, but we think that defining capability-specific games and then reporting the "game score" should serve equivalent effect.
>
> W2 - Apologies about the potential misunderstanding. Do you mean: “Using a chain-of-thought prompt?” Yes, as explained in the general rebuttal (The SmartPlay Interface section), SmartPlay is compatible with any prompting frameworks. In addition, the experiments we conduct (page 7, Section 4) uses the “let’s think step by step” prompt to promote explicit reasoning for all LLMs. Because of the explicit reasoning prompt, we were able to make qualitative observation that the LLMs are confused with the Tower of Hanoi game in section 4.2

---

### Author Response · Authors · 2023-11-11

We thank all the reviewers for their insightful comments. We are encouraged by all reviewers' appreciation that our benchmark is original and high-quality (Reviewers itPd, Jri2). In addition, we are encouraged by all reviewers’ acknowledgements that SmartPlay 1) addresses a critical aspect of LLM evaluation using a 2) comprehensive range of games.

The SmartPlay Interface
We would like to emphasize that SmartPlay is a unified game API that, at its core, outputs text observation and expects an action (integer) as input (Section 3). Therefore, SmartPlay runs independently from any agent architecture, and therefore could also be used as a prompt evaluator.
We note that the codebase is modular and could be easily extended by adding more games into SmartPlay. All games with text observation descriptor and a clear score/reward function can be easily integrated into SmartPlay.

---

> ### Author Response · Authors · 2023-11-14
> **Edits to the manuscript**
>
> -- Nov 17, 2023 --
> 1. We added Section B in the appendix to provide implementation detail and example of how the visual description for Minecraft is generated.
>
> -- Nov 14, 2023 --
> 1. We added 2 new subsections (Section C.1, C.2) in the appendix to give detail on collection of human baseline scores (C1), and provide the formula we used to calculate normalized human score (C2).
>
> -- Nov 13, 2023 --
> 1. We added a new section (Section 2) to formally explain our choice of the 9 capabilities and define the degrees of challenges
> 2. We re-phrased section 3.1 and Figure 2-left table for better clarity.

---

### Author Response · Authors · 2023-11-13
**GitHub has been updated with code**

As requested by the reviewers, we have uploaded full code base to the anonymous repository documented in the submission (github.com/LLMsmartplay/SmartPlay).

Please be advised that this repository is for anonymous review ONLY and will not receive any updates. We will de-anonymize the link to an actively maintained repo in the final version of our paper.

---

### Comment · Area_Chair_CPG8 · 2023-11-20
**Please engage in reviewer-author discussions**

Reviewers - I encourage you to read the authors' response carefully and let the authors know whether their response has addressed your comments.

---

### Meta-Review · Area_Chair_CPG8 · 2023-12-03

**Metareview:**

The paper introduces a novel benchmark consisting of six games designed to assess the capabilities of LLMs as intelligent agents. These games offer unique challenges, including long-horizon planning, understanding the odds, spatial reasoning, and learning from past interactions. SmartPlay addresses the gap in systematic evaluation of LLMs beyond language-based tasks and allows for specific analysis of various LLM capabilities. By evaluating across the diverse tasks, the benchmark provides insights into the current state of LLMs and identifies key areas for further development towards next-generation intelligent agents. The reviewers generally appreciate the benchmark design quality, the paper clarity, and the contribution significance. The concerns initially raised by the reviewers (how the visual description for Minecraft is generated, how to determine difficulties, etc) were sufficiently addressed. Thus, the AC recommends accepting this paper.

**Justification For Why Not Higher Score:**

I'd expect that a spotlight paper would be one that brings more significant breakthrough or insight to the community. As reviewers point out, the proposed games might not be challenging enough for LLMs, as GPT-4 aces already most of them besides crafter and Minecraft.

**Justification For Why Not Lower Score:**

The paper proposed a suite of well-defined benchmarks with a clear documentation, which can be very beneficial to the community.

---

### Decision · Program_Chairs · 2024-01-16

Accept (poster)